# Career orientations of medical students: A Q-methodology study

**Lokke Gennissen**[1]*, **Karen Stegers-Jager**[1], **Job van Exel**[2,3], **Lia Fluit**[4], **Jacqueline de Graaf**[4,5], **Matthijs de Hoog**[1,6]

1 Institute of Medical Education Research Rotterdam, Erasmus Medical Center Rotterdam, Rotterdam, The Netherlands, 2 Erasmus School of Health Policy & Management, Erasmus University Rotterdam, Rotterdam, The Netherlands, 3 Erasmus School of Economics, Erasmus University Rotterdam, Rotterdam, The Netherlands, 4 Radboudumc Health Academy, Radboud University Medical Center Nijmegen, Nijmegen, The Netherlands, 5 Department of Internal Medicine, Radboud University Medical Center Nijmegen, Nijmegen, The Netherlands, 6 Department of Pediatrics, Erasmus Medical Center Rotterdam, Rotterdam, The Netherlands

* L.Gennissen@erasmusmc.nl

**Data Availability Statement:** For the sake of confidentiality, full data are not publicly accessable. Additional anonymized data is available upon request. Data available on request are the Q-sort data and the anonymized interviewdata (in Dutch).

## Abstract

### Introduction

In pursuing optimal health care, an adequate medical workforce is crucial. However, many countries are struggling with a misalignment of students' specialty preferences and societal needs regarding the future medical workforce. In order to bridge this gap, it is relevant to gain a better understanding of the medical career choice processes. We explored career orientations among medical students in the Netherlands and their implications for future career choices.

### Methods

We used Q-methodology, a hybrid qualitative–quantitative method, to explore career orientations of medical students. Medical students from two universities in the Netherlands, varying in year of progression of medical school, ranked 62 statements with regard to importance for their future career choice. Participants explained their ranking in an interview and completed a questionnaire regarding demographics. Using by-person factor analysis we identified groups of individuals with similar orientations.

### Results

Twenty-four students participated in this study, resulting in three distinct orientations towards future careers: a first career orientation that highly values lifelong self-development; a second that values work-life balance, and a third that was more concerned with achievement and recognition of their work.

### Conclusion

Medical students' career orientations differed in the importance of challenge, work-life balance, and need for recognition. This knowledge can help to design interventions to shift

Data requests can be sent to Prof. Dr. Walter van den Broek, scientific director of the institute of Medical Education Research Rotterdam (W.W. vandenbroek@erasmusmc.nl).

**Funding:** The research program is funded by the Dutch Ministry of Health, Welfare and Sports— Project: Dedicated Schakeljaar. The funding body had no influence on the design of the study and collection, analysis, and interpretation of data and writing of the manuscript.

**Competing interests:** The authors have declared that no competing interests exist.

career choices of medical students closer towards future needs in society. Offering career coaching to students that challenges them to explore and prioritise their values, needs and motivations, for example using the materials form this study as a tool, and stimulates them to consider specialties accordingly, could be a promising strategy for guiding students to more long-term satisfying careers.

## Introduction

### Mismatch

As society is rapidly changing, health care demands are changing as well. Matching the supply of health professionals to societal demands is essential in aspiring to maintain an affordable healthcare system, which is sustainable and fit for purpose [1]. However, in many Western societies there is a continuing imbalance between the career aspirations and paths of medical doctors and societal needs with regard to the professionals required for optimal care delivery [1, 2]. Although there is no shortage of medical students, the distribution of medical graduates over specialties is problematic. Whereas society needs for instance more public health and elderly care doctors, medical graduates rather pursue careers in pediatrics or internal medicine [1, 3]. This mismatch has led to both shortages and unemployment of medical professionals. Solving this mismatch is of urgent interest for the medical workforce as well as society. For that purpose, a better understanding of the career aspiration of medical students is essential.

### Career aspirations

The maldistribution of students in terms of specialties has been increasingly addressed by researchers [1, 4, 5]. Career aspirations of medical students appear to lie at the heart of the problem, as they play an important role in career decisions such as medical specialty choice. Students often already enter medical school with certain ambitions, concerns, and hopes about the medical field. Ideally, through experiences in the field during medical school, students gradually learn to realise what they need and like, what they more deeply believe or value about work and life, what they are good at, and what skills and abilities are critical. These motives, values and talents gradually come together in a comprehensive pattern of job-related preferences that will drive their aspirations, which is also referred to as their career orientation. Insight in the career orientations of medical students will likely help us to better understand the drives in their medical specialty choice process. This knowledge is essential when striving to address issues and propose solutions for the problematic distribution of medical graduates over specialties.

The process of how medical students develop their career orientation and how they come to their medical specialty preferences is complex and has been researched for many years [6–8]. In general, scholars have tried to identify aspects of importance for medical specialty choice in students' characteristics (such as gender and personality) [9], values (e.g. personal preference) [10], career needs or preferences (e.g. expected income or working hours) [11], perceptions of medical specialty characteristics (e.g. personal experience) [12], and medical curricula characteristics (e.g. curriculum design) [13]. Although this has resulted in increased knowledge on a broad variety of aspects which play a role in the medical specialty choice process, a clear understanding of the process is still lacking [8].

The present study sought to contribute to the understanding of this medical specialty choice by approaching it as a decision-making process in which all relevant aspects of the choice are interrelated. Cleland and colleagues have preceded us in taking this perspective and argued that instead of looking at individual aspects one could also focus on push and pull factors [14, 15], the principal aspects that either drive people away from a certain alternative or draw people towards it. Their studies, in which they used discrete choice experiments, probably come closest to investigating how medical graduates in the UK value and trade-off different aspects in their career decision-making. This led to insights regarding the importance of these aspects relative to financial gains.

Such choice experiments, by design, are suitable for investigating the relative importance of a limited number of aspects. However, a large and varied set of values, needs and motives may play a role in specialty choices of medical students. Therefore, a broader understanding of how all these different aspects are prioritised by medical students in their speciality choices is essential for designing interventions that could help solve the maldistribution of the medical workforce. In addition, one may expect considerable heterogeneity between students in what they consider important for their future careers. Thus, in our study we investigated how students trade-off all the relevant aspects with regard to medical specialty choice using Q-methodology. This approach makes it possible to ask participants to rank an extensive set of items, and to identify similarities as well as differences between distinct career orientations of medical students. In addition, by collecting both quantitative and qualitative data, rich interpretations of different career orientations can be generated.

Although Q-methodology has previously been used in career research [16–18], this study is the first to apply this approach to medical career research.

In most career choice theories, the career choice process is approached mainly as a matching process. This means that it is assumed that a person -consciously or unconsciously- prefers a specialty that matches their individual needs, values, motives and talents [19]. Consequently, self-knowledge about what you want and what your qualities and values are, is essential for making appropriate career choices. Many of these theories thus suggest that starting with a thorough self-exploration is necessary for medical students to develop a career orientation and for effective career decision-making. The latter is all the more relevant because choosing the proper medical specialty will benefit job satisfaction, and thereby the quality of care delivered [19–22]. But even more important in the context of the current study, more insight into the career orientations of medical students could improve our understanding of their medical specialty choices [19].

### Research aim

This study aims to contribute to the understanding of the process of medical specialty preference formation, by focusing on student career orientations. Using Q-methodology [23], this study explores patterns in the importance medical students attach to a broad set of values, needs, and motives for specialty choice, and uses quantitative and qualitative data to describe these patterns as distinct career orientations. Finally, the study elaborates on their implications for medical specialty choices and potential solutions for the mismatch between the aspirations of medical students and societal needs for medical professionals.

## Methods

### Study design

The mixed method characteristic of Q-methodology is particularly suited to systematically explore and explain patterns and diversity in a subjective phenomenon [23]. Data are gathered

by asking the participants to rank a set of statements that represent the broad set of issues that are relevant to the topic of study. The assumption is that participants reveal their viewpoint through ordering the full statements set, and that if different participants rank the statements in a similar way, they hold a similar view on the topic. To identify similarities and differences between participants, the ranking data is subjected to by-person factor analysis, thus correlating the persons instead of the statements [23].

## Context

This study took place in medical schools at two universities in the Netherlands. Demographic characteristics of their students and their curricula are quite similar [24]. Both universities are metropolitan universities, with similar distribution of background characteristics of the students. Both universities have a 6-year undergraduate medical curriculum of which the 3-year Bachelor is mainly theoretical, and the 3-year Master is mainly clinical. In the Netherlands, medical school graduates that aim to become a medical specialist have two options: one can apply for a medical residency selection procedure directly after medical school or one can start working in a specialty as a resident-not-in-training. The latter option is frequently used to get a better sense of whether a specialty fits, or to gain work experience to build a better resume before applying for the specialty of choice. As a consequence, the moment of the actual medical specialty decision making can differ between individual students.

## Participants

Participants in this study were students from all six years of the undergraduate curriculum of the two medical schools. As we had no prior information about which career orientations exist among students and which students have these different orientations, a combination of convenience sampling and snowballing was used. As a starting point of the data collection we approached a convenience sample of medical students, i.e. an easy-to-reach group consisting of participants varying in year of medical school and background characteristics that we anticipated to be relevant for capturing the diversity of viewpoints among medical students, as is common in Q-methodology [23]. In order to reach a more diverse group, snowballing was used. This means we asked participants from the convenience sample to suggest consecutive participants, based on their expectations that these other students would have similar or dissimilar career aspirations. The main purpose of our sampling strategy was including a diverse group of medical students that would help us to identify the variety in career orientations. Since participants in a Q-methodology study evaluate and rank a large number of items, a limited number of well-selected participants is sufficient to reach saturation and to identify the main viewpoints about a topic [23]. All participants completed the study.

## Statement set

A crucial part in conducting a Q-methodology study is developing a comprehensive set of statements, covering the variety of things that people may say or think about the issue being investigated, allowing all participants to reveal their viewpoint [23]. Therefore, a team of clinical and medical education professionals joined forces to develop the statement set. We aimed the set to represent the diversity of possible responses to the question: "thinking about your future career, what is important to you in your medical specialty choice?". To start, aspects important in specialty preference were identified in the literature. The long-list of items retrieved was reviewed and supplemented with experiences of the team. Next, these items were transformed into statements that could be an answer to the previously stated question. For example: "Working in a prestigious medical specialty". Subsequently, the statements were

discussed with medical students and specialists regarding their ambiguity, clarity, and suitability for use with medical students. After adjustments, the statement set was pilot-tested among students. Pilot participants were asked to comment on the completeness and comprehensibility of the set of statements. Final revisions (by JvE, KSJ and LG) resulted in a set of 62 statements. Themes covered in the statement set included: Personal development/advancement, Lifestyle, Prestige, Service/Altruism, Security and stability, Autonomy and Work-related content.

## Data collection

Participants were interviewed in person (by LG). The ranking was instructed as follows: first, participants were handed the statements printed on cards, in random order, asked to read all statements and to categorise them in three piles: important, unimportant and neutral for their choice of a medical specialty. Next, they were asked to take the statements in the "important" pile, to read them once again, and to select the two statements they felt were the most important for their choice of a medical specialty, and place these on the right-hand side of the score sheet in the two spots below the number 5 (see Fig 1). Next, they were asked to read through the remaining statements of the "important" pile, select the four statements they now felt were the most important, and place them in the next column, below 4. This process was repeated until no statements remained in the pile. Then, a similar procedure was followed for the statements in the "unimportant" pile (then ordering from left to right), and for the "neutral" pile (placed in the remaining spots in the middle). These sorting phases are shown in Fig 2. After finishing the ranking of the statements, participants were asked in a face-to-face interview to explain their ranking and in particular the placement of the statements at the extreme ends, and to answer some demographic background questions in a paper-based questionnaire (see S1 Appendix).

## Analysis

By-person factor analysis was used to identify distinct career orientations from the individual ranking data [23]. First, a correlation matrix between the rankings of the statements by the participants is computed. Assuming that if two participants rank the 62 statements in a similar way, they have a similar career orientation, factor analysis (i.e., centroid factor extraction followed by varimax rotation [23]) is then applied to identify the main patterns in the ranking data. The selection of the number of patterns (or factors) to extract from the data is based on statistical criteria (i.e., Eigenvalue >1 and a minimum of two participants statistically significantly associated with the pattern) and whether patterns have a coherent interpretation that is also supported by the corresponding qualitative data from the interviews.

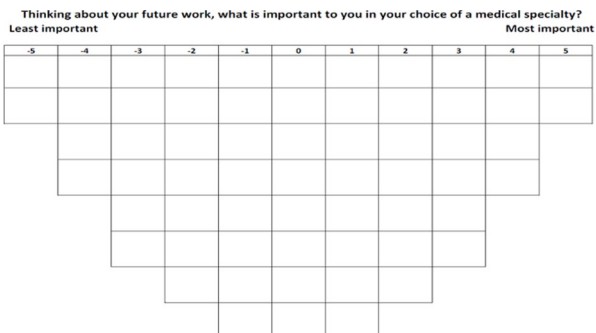

**Fig 1. Score sheet.**

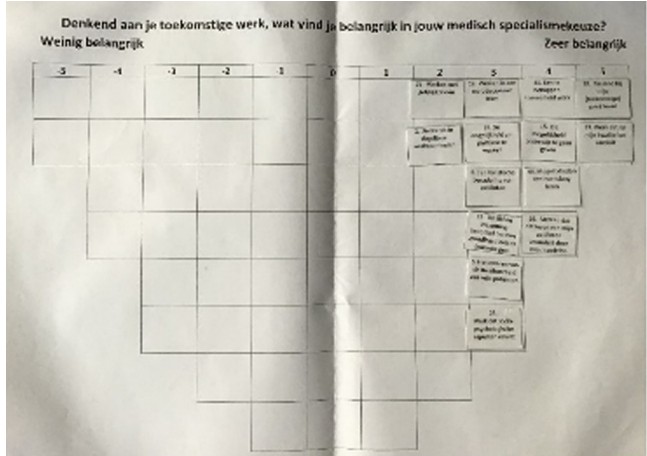
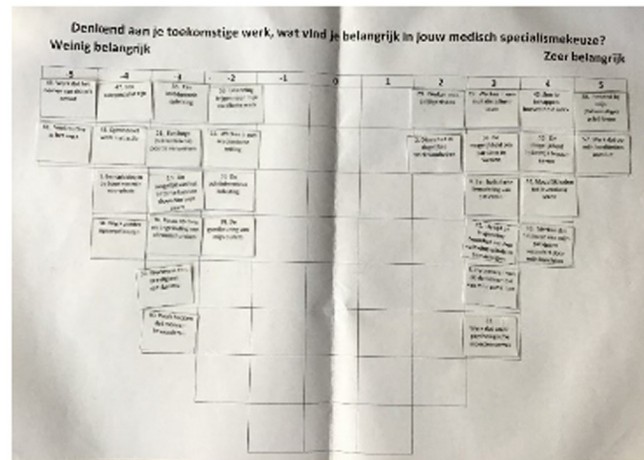
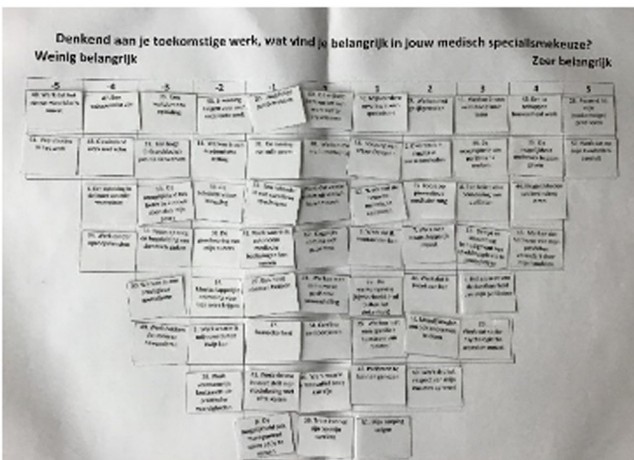

**Fig 2. Step-wise representation of the sorting process.**

For each identified pattern, an idealized ranking of the data is computed. This is a weighted average ranking of the 62 statements, based on the rankings of participants statistically significantly associated with the pattern (p < .0.05) and their correlation coefficient as weight.

These idealized rankings are then interpreted as distinct career orientations among medical students. Interpretation of each pattern starts by the characterising statements, those with a +5, +4, -4 or -5 score in that pattern, and the distinguishing statements, those with a statistically significant different score (p<0.05) in a pattern as compared to the other patterns. However, a pattern consists of all 62 statements and it is the interrelationship of the many items that ultimately drive our interpretation.

Finally, the qualitative interview data (i.e. the motivations for their ranking of the statements provided by participants) from participants statistically significantly associated with the pattern (p < .0.05) are used to verify and refine the interpretations. Selected quotes from these qualitative materials are used to substantiate the interpretations of the patterns.

## Ethical considerations

Prior to participation, all participants were informed about the purpose of the study. Informed consent was obtained beforehand. Participation was voluntary. Participants were explicitly

informed that this research did not influence their progression at the medical course. All participants gave written informed consent. Data were processed confidentially and were handled according to the requirements of the Dutch data protection authority. This study was approved in the Netherlands by the NVMO Ethical Review Board (NERB number 812).

## Results

Saturation was reached after 24 interviews with medical students. During the last few interviews, no really distinct rankings of the statements or novel motivations for ranking the statements were found. These twenty-four students were from medical schools at two universities, 9 were male and 15 were female. The mean age of participants was 21.3, with a range of 18–27 years. This gender and age distribution is representative of the total student populations [25]. Five participants were in the first year of medical school, three of them in the second year, four in the third year, four in the fourth year, four in the fifth year and three in the sixth year. Table 1 presents the demographic data of our participants.

The 24 rankings of the statements supported a maximum of three patterns (i.e. career orientations), which, after inspection for clarity of interpretation and distinctiveness, were retained for interpretation. These three patterns together explained 48% of the total variance in the ranking data, and between 5 and 10 participants were statistically significantly associated with each pattern. These career orientations will be referred to as life-long learning as a calling, work-life balance and achievement and recognition.

**Table 1. Demographic characteristics participants.**

| # | Study year | Gender | Nationality | First generation higher education | Age (in years) |
|---|---|---|---|---|---|
| 1 | 4th | Male | Dutch | No | 20 |
| 2 | 4th | Female | Dutch | No | 21 |
| 3 | 3rd | Female | Dutch | Yes | 20 |
| 4 | 3rd | Female | Dutch | No | 20 |
| 5 | 2nd | Male | Non-Western | No | 19 |
| 6 | 1st | Female | Dutch | No | 18 |
| 7 | 3rd | Female | Western | No | 20 |
| 8 | 2nd | Female | Dutch | No | 19 |
| 9 | 5th | Male | Dutch | No | 22 |
| 10 | 5th | Male | Dutch | No | 23 |
| 11 | 4th | Male | Dutch | No | 24 |
| 12 | 6th | Female | Dutch | No | 27 |
| 13 | 1st | Female | Dutch | No | 18 |
| 14 | 1st | Female | Dutch | No | 22 |
| 15 | 1st | Female | Non-western | No | 19 |
| 16 | 1st | Female | Dutch | No | 18 |
| 17 | 5th | Female | Western | No | 24 |
| 18 | 3th | Female | Non-Western | No | 20 |
| 19 | 5th/6th | Female | Dutch | No | 25 |
| 20 | 5th | Female | Dutch | No | 25 |
| 21 | 4th | Male | Dutch | No | 22 |
| 22 | 6th | Male | Dutch | No | 24 |
| 23 | 3rd | Male | Non-Western | No | 22 |
| 24 | 2nd | Male | Non-Western | No | 21 |

Table 2 presents the idealised ranking of the 62 statements for the three patterns, highlighting with an asterisk the statements that were identified as distinguishing statements for these patterns.

In the next section we present the interpretations of the three patterns as distinct career orientations of medical students. Between parentheses, the scores of the statements for that particular career orientation are given, with the numbers ranging from -5 to +5, corresponding to the placement of the statement on the grid (see Fig 1). Additionally, several quotes given by participants who loaded onto the particular career orientation are provided as support for our interpretation, recognisable by the use of italics for these quotes. Quotes are identifiable by an "R" followed by a number referring to the unique identifier of the respondent. Tables 3–5 show the characterising statements of the three career orientations.

## Career orientation 1: Lifelong learning as a calling

This career orientation was represented by the Q-sorts of three male and three female participants, being first, second, third and fourth year medical students. Students with this career orientation place high value on opportunities to develop oneself (st.44: +4): "*That is what I love to do. Keep learning. So that is also what I'm looking for in a future job*" [R5]. They are interested in getting involved in research (st.55: +4), being innovative (st.10: +2) and perhaps taking up a teaching role (st.15: +2), and therefore find the academic work setting appealing (st.11: +2). They also place high value on being able to physically and mentally handle their work (st.6: +5; st.46: +4): "*But it shouldn't make you unhappy, you should be able to cope with it mentally*" [R16]. An important feature of their future work is its impact on the lives of patients (st.16: +5) and, most of all participants, they seek a setting where they would be able to have longer relationships with patients (st.20:+2; st.36: 0). They don't pursue work with excitement and action and also attach little importance to the diversity in daily tasks (st.2: +3; st.43:-1).

These medical students mostly want to follow their calling (st.51: +4). Accordingly, they are concerned less than others with external recognition for their job (st.38: -5; st.49: -1; st.21: -4; st.50: -4; st.40: -3; st.14: -2; st.13: -4), work-life balance (st.28:+1), or the possibility the job provides to pay off their study debt swiftly (st.41: -5): "*Look, that loan has really quite favourable terms, relatively, there's no hurry as such and yes sooner or later it'll come to it, but there's no hurry*" [R5].

## Career orientation 2: Work-life balance

This career orientation was represented by 10 participants, nine female and one male, and they varied from second to sixth year medical student. Although medical students with this career orientation also find it important to follow their calling (st.51: +2), they most of all emphasise aspects that are associated with work-life balance. To match their future job with their ideas of their private life is the most important aspect to them. (st.28; +5): "*Later on I would just like to be happy at home and not only at work, so it is important to me that the two go well together*" [R8]. Accordingly, they attach high importance to job aspects like distance to work (st.1: 0), time and effort needed for obtaining a training position (st.12: +2), and opportunities to work part-time (st.39: +2).

They also prefer a manageable workload (st.45: +3) and, least of all participants, seem to mind routine in work (st.61: -1). Another aspect of work they really care about is whether the work suits their qualities (st.57:+5). They are not very interested in taking up a management or leadership role (st.9: -2; st.21: -4), or being innovative (st.10: 0) or working with the newest technical instruments (st.32: -5).

**Table 2. Idealised rankings of the statement.**

| # | Statement | Pattern 1: Life-long learning as a calling | Pattern 2: Work-life balance | Pattern 3: Achievement and recognition |
|---|---|---|---|---|
| 1 | Residency training available near my current home town | -3 | 0** | -3 |
| 2 | Diversity in daily tasks | 3** | 4 | 5 |
| 3 | Work in which I can be creative | 1 | 2 | 1 |
| 4 | Work that is mostly about clinical reasoning | 0 | 1 | 2 |
| 5 | To experience the gratitude of my patients | 2 | 2 | 3 |
| 6 | Work that I can handle mentally | 5** | 3 | 2 |
| 7 | Work that has societal impact | 0 | -1 | 3** |
| 8 | Having a holistic approach to patients | 0 | 0 | 0 |
| 9 | Opportunity to take up a management role | 0 | -2* | 1 |
| 10 | Work in which I can be innovative | 2* | 0 | 1 |
| 11 | Working in academic medicine | 2** | 0* | -2* |
| 12 | The time and effort needed for getting a training position | -2 | 2** | -4 |
| 13 | Possibility to do better than my peers | -4 | -4 | -1** |
| 14 | Gaining societal recognition for my work | -2 | -2 | 1** |
| 15 | Opportunities to take a teaching role | 2 | 0 | 0 |
| 16 | To experience the impact my work has on the life of my patients | 5 | 4 | 3 |
| 17 | Having security of employment | 2 | 3 | 0* |
| 18 | Focus on one organ or organ system | -2 | -4 | -4 |
| 19 | Working in a multidisciplinary team | 1 | 3 | 3 |
| 20 | Work where I can have a longer relationship with patients | 2* | 0 | 1 |
| 21 | Obtaining a high hierarchical position | -4 | -4 | -1** |
| 22 | The setting (e.g. inside/outside a hospital) where I work | 3 | 1 | 2 |
| 23 | Work that involves sociopsychological aspects | 2 | 1 | -1** |
| 24 | Working with a diverse patient group | 1 | 1 | 3 |
| 25 | Working with a specific spectrum of diseases | -2 | -1* | -3 |
| 26 | Work that is mostly about performing practical skills | -1 | -2 | -2 |
| 27 | Earning a high income | -2 | -3 | -2 |
| 28 | Compatibility with my (future) private life | 1** | 5 | 4 |
| 29 | Working with like-minded people | 1 | 2 | 0 |
| 30 | Working with multi-morbidity | -1 | -1 | -2 |
| 31 | The opinion of my peers | -1 | -3 | -3 |
| 32 | Working with newest technical equipment | -2** | -5 | -4 |
| 33 | A role model I have (had) in a specific specialty | -3 | -2 | -1 |
| 34 | Work where I do not have to be on call | -3 | -1 | -2 |
| 35 | A short residency training program | -4 | -3 | -5 |
| 36 | Focus on guiding chronically ill patients | 0** | -3 | -2 |
| 37 | Focus on preventive medicine | 0 | -2 | -1 |
| 38 | My parents' approval | -5 | -4 | 0** |
| 39 | Being able to work part-time | -3 | 2** | -3 |
| 40 | Having work that people admire | -3 | -3 | 1** |
| 41 | Work that enables me to pay off my study loans quickly | -5 | -3* | -4 |
| 42 | Being able to cure people | 3 | 3 | 3 |
| 43 | Work which involves excitement and action | -1* | 1* | 4** |
| 44 | Opportunities for lifelong learning | 4** | 1 | 2 |

(*Continued*)

**Table 2.** (Continued)

| # | Statement | Pattern 1: Life-long learning as a calling | Pattern 2: Work-life balance | Pattern 3: Achievement and recognition |
|---|---|---|---|---|
| 45 | Having a manageable workload | 1 | 3** | -1 |
| 46 | Work that I can handle physically | 4** | 2 | 1 |
| 47 | To be a subspecialist | -1 | -1 | -3 |
| 48 | Work in which I am required to take risks | 0 | -1 | 0 |
| 49 | Work that is respected by the people close to me | -1 | -2** | 0 |
| 50 | Working in a prestigious medical specialty | -4 | -5 | -3 |
| 51 | Following my calling | 4 | 4 | 2** |
| 52 | Work where I have autonomy in making medical decisions | -1 | 0 | 1 |
| 53 | Experience I have in a specific specialty | 0 | 1 | -1** |
| 54 | Opportunities for career advancement | 1 | 2 | 2 |
| 55 | Opportunities to get involved in research | 4** | 0* | -2* |
| 56 | The amount of administrative work | -2 | -1 | -1 |
| 57 | Work that suits my qualities | 3* | 5 | 5 |
| 58 | Being recognised for my excellence | -1 | -2 | 0 |
| 59 | To feel proud of my career achievement | 3 | 3 | 4* |
| 60 | Having freedom to organise my own work | 1 | 1 | 2 |
| 61 | Work that includes much routine work | -3* | -1** | -5* |
| 62 | Having daily contact with patients | 3* | 4 | 4 |

For each statement for each pattern a number ranging from -5 to +5 is displayed. This corresponds to the location of the statement on the grid (as is shown in Fig 1) in the idealized ranking representing that pattern. An asterisk identifies a distinguishing statement.

\* = $p < 0.05$;

\*\* = $p < 0.01$

**Table 3. Characterising statements pattern 1.**

| # | Statement | Pattern 1: Life-long learning as a calling | Pattern 2: Work-life balance | Pattern 3: Achievement and recognition |
|---|---|---|---|---|
| 6 | Work that I can handle mentally | 5** | 3 | 2 |
| 16 | To experience the impact my work has on the life of my patients | 5 | 4 | 3 |
| 44 | Opportunities for lifelong learning | 4** | 1 | 2 |
| 46 | Work that I can handle physically | 4** | 2 | 1 |
| 55 | Opportunities to get involved in research | 4** | 0* | -2* |
| 51 | Following my calling | 4 | 4 | 2** |
| 13 | Possibility to do better than my peers | -4 | -4 | -1** |
| 21 | Obtaining a high hierarchical position | -4 | -4 | -1** |
| 35 | A short residency training program | -4 | -3 | -5 |
| 50 | Working in a prestigious medical specialty | -4 | -5 | -3 |
| 38 | My parents' approval | -5 | -4 | 0** |
| 41 | Work that enables me to pay off my study loans quickly | -5 | -3* | -4 |

**Table 4. Characterising statements pattern 2.**

| # | Statement | Pattern 1: Life-long learning as a calling | Pattern 2: Work-life balance | Pattern 3: Achievement and recognition |
|---|---|---|---|---|
| 28 | Compatibility with my (future) private life | 1** | 5 | 4 |
| 57 | Work that suits my qualities | 3* | 5 | 5 |
| 2 | Diversity in daily tasks | 3** | 4 | 5 |
| 16 | To experience the impact my work has on the life of my patients | 5 | 4 | 3 |
| 51 | Following my calling | 4 | 4 | 2** |
| 62 | Having daily contact with patients | 3* | 4 | 4 |
| 13 | Possibility to do better than my peers | -4 | -4 | -1** |
| 18 | Focus on one organ or organ system | -2 | -4 | -4 |
| 21 | Obtaining a high hierarchical position | -4 | -4 | -1** |
| 38 | My parents' approval | -5 | -4 | 0** |
| 32 | Working with newest technical equipment | -2** | -5 | -4 |
| 50 | Working in a prestigious medical specialty | -4 | -5 | -3 |

Financial and job security seem to be of greater importance to students in this viewpoint when compared to the other patterns (41:-3; 17: +3): "…*at the end of the day you want to do an education you can actually use and ultimately gets you a job*" [R7].

Like medical students with the first career orientation, they perceive external recognition with regard to their job as of little importance (st.38: -4; st.49: -2; st.21: -4; st.50: -5; st.40: -3; st.14: -2; st.31: -3).

**Table 5. Characterising statements pattern 3.**

| # | Statement | Pattern 1: Life-long learning as a calling | Pattern 2: Work-life balance | Pattern 3: Achievement and recognition |
|---|---|---|---|---|
| 2 | Diversity in daily tasks | 3** | 4 | 5 |
| 57 | Work that suits my qualities | 3* | 5 | 5 |
| 43 | Work which involves excitement and action | -1* | 1* | 4** |
| 59 | To feel proud of my career achievement | 3 | 3 | 4* |
| 28 | Compatibility with my (future) private life | 1** | 5 | 4 |
| 62 | Having daily contact with patients | 3* | 4 | 4 |
| 12 | The time and effort needed for getting a training position | -2 | 2** | -4 |
| 18 | Focus on one organ or organ system | -2 | -4 | -4 |
| 32 | Working with newest technical equipment | -2** | -5 | -4 |
| 41 | Work that enables me to pay off my study loans quickly | -5 | -3* | -4 |
| 35 | A short residency training program | -4 | -3 | -5 |
| 61 | Work that includes much routine work | -3* | -1** | -5* |

### Career orientation 3: Achievement and recognition

This career orientation was represented by the sorts of three male and two female students, which were first, fourth and fifth year medical students. In this career orientation it is less about following a calling (st.51: +2) but more about finding a job that fits with their qualities (st.57: +5) and will provide sufficient diversity (st.2: +5) and excitement (st.43: +4) in day-to-day work. These students do not care for much routine work (st.61:-5): "*Well, you know, what strikes me is something I don't care about, that it is really routine and quite opposite to what I do care about, that it is diverse and exciting as well*" [R10].

For these medical students it is important that they have societal impact (st.7: +3) and can be proud of their achievements (st.59: +4): "*Be proud of my career, just feel good about it. Even if it wouldn't be in medicine, I just want to enjoy the work I do, be proud of it, be good at it, 'cause that is what would give me satisfaction*" [R9]. However, it is also important that they gain recognition (st.14:+1; st.58: 0) and admiration (st.40: +1) from others; their parents and people close to them(st.38: 0; st.49: 0), but also from society (st.14: +1). More than others, these medical students appreciate job autonomy (st.60: +2; st.52; +1), would not mind to outperform their peers (st.13: -1) and have some interest in taking up a management or leadership role (st.9: +1; st.21: -1). Less than others they aim for work in academic medicine (st.11: -2) and they are little concerned with aspects like the time and effort needed for getting in a training position (st.12: -4), the location of the residency (st.1: -3), the possibility to work part-time (st.39: -3), job security (st.17:0) or a manageable workload (st.45: -1). Of least interest to them was a short residency training program (st.35: -5).

## Discussion

In this study, we identified three distinct career orientations among medical students in the Netherlands: one very much focused on lifelong self-development; a second focused more on work-life balance; and a third which was more concerned with achievement and recognition of their work.

The patterns varied mostly on features which were not per se medical content related. Differences emphasised socio-economic and occupational feature preferences, such as lifestyle, prestige and the need for challenge. Students with a "lifelong learning as a calling" orientation expressed a need to be challenged and a desire to have a career that provides them opportunities to keep learning and developing. These students appear willing to sacrifice lifestyle features to a certain extent, yet are not willing to go beyond their physical and mental limits. Additional features of importance were having an impact on the lives of patients with their work and having the feeling to follow their calling. Recognition from others seems to be of minor importance. Medical students with a "work-life balance" orientation to their career primarily expressed a desire for a good balance between work and private life. They expressed lower challenge needs and lowly valued recognition. The "achievement and recognition" career orientation sets itself apart from the other two by a relatively stronger emphasis on recognition. Responsibility and autonomy are distinctively more valued compared to the other two patterns. They are willing to trade-off lifestyle features to some extent in return for diversity of daily tasks, a job fitting with their qualities, recognition and impact.

The primary differentiating elements in career orientations thus seem to be need for challenge, work-life balance and recognition. Especially work-life balance is receiving growing attention in recent literature, reflecting its importance for the current generation [26]. This is an interesting finding since it contrasts with the fact that the medical profession is known to be a demanding profession in which physicians work hard, and make long days. Remarkably, particularly female participants defined the career orientation work-life balance. This

methodology is not suited to make definitive statements about prevalence of a career orientation in certain subgroups. However, one might wonder whether work-life considerations might still be of more importance to female medical students. Notwithstanding, recent research showing the importance of work-life balance for male students [27], our data suggest that female participants give a higher priority to work-life considerations. Students in the work-life balance also were further along their education. This prompts the question whether more experience in the field will push students to the work-life orientation or that life-events along the way will make students turn to this career orientation. The current generation of medical specialists tend to place work first, while the current generation of medical students and residents (often referred to as millennials) [28] might value time off and lifestyle more [29]. Open conversations on lifestyle preferences and career choices between medical students or graduates and the current generation of medical specialists might be hindered by the general expectations of the latter that one should prioritise work over private life [30]. These expectations might convey (unconscious) messages to students, raising barriers to discuss work-life balance features with their educators and role models. This is even more relevant as work-life balance is one of the major reasons for attrition during residency and career changes of physicians already in practice [31–34]. Therefore it is important to consider the preferences of medical students with regard to work-life balance in medical training and work settings. A complicating aspect for medical students in taking work-life balance values into account in their medical specialty choice is that their work-life balance values might shift over time, making it difficult to make a lifelong sustainable specialty choice. However, it is expected that reflecting on and exploring one's career decisions can help individuals to make decisions that are compatible with their current lifestyle and flexible to changes therein, in order to meet with evolving work-life needs during their career [19].

In the interviews following the ranking of the 62 statements, students were asked how they experienced the sorting exercise. All participants recognised that prioritising their values, needs, and motivations in this particular way helped them to get insight in their priorities. In their opinion this would help them in thorough career decision making, thereby suggesting that Q-sorting could be used as a career guidance tool. This also creates opportunities to address the maldistribution by on one hand opening a new window on medical specialties to become more attractive, while on the other hand guiding students to the insight what features of future work are most important to them and stimulating them to explore the medical specialties accordingly. Many features (e.g. "working in a multidisciplinary team") can be found in multiple specialties. Showing this to students by enabling a broad exploration might be a promising start in striving towards a better distribution over the specialties.

Because the approach used in this study stimulated participants to consider and prioritise a broad range of aspects that may be relevant to their career orientation, our results offer an in-depth understanding of important features guiding specialty preferences among medical students. It is anticipated that by exploring and prioritising their values, needs and motivations in this way, students are stimulated to go through their career decision making process in a more rational way. Although this could be a way to improve the quality of the decision [35], it is probably not how most students approach this process in practice. If so, students could benefit from career counselling, or individual coaching.

It is tempting to link a specific career orientations to certain medical specialties. Although this would certainly be interesting to further explore in future research, there is a pitfall to it. First of all, this study showed us the work-related content statements were not the distinguishing factors between these career orientations. That raises the question whether different career orientations can be found in medical specialists in a medical specialty. Secondly, linking

certain career orientations to medical specialties might inhibit the exploration phase, risking reinforcement of the maldistribution over the medical specialties.

Some limitations of this study need to be highlighted. Although our sample strategy was focused on maximising variety in order to collect the potentially wide range of perspectives, it is possible that we have missed an important career orientation among medical students. Beforehand we expected to find more heterogeneity in career orientations. This lack in heterogeneity might be explained either by our study design, or by a homogeneity in medical school. Regarding our study design, we might have sampled from a homogeneous group of students, despite our attempt to actively seek maximum variety. However, this homogeneity might also be caused by medical school selection, perhaps selecting students with certain motives, values or talents, or it might be that there is a wider variety in career orientations at the start of medical school, but that through learning or socialisation (i.e. students learning a way of perceiving, thinking and acting) this variety decreases over the years.

Therefore a promising route for future research would be to explore if, how and why career orientations change during medical school, and whether this differs between students and schools with different characteristics. This might be particularly valuable in the context of increasing diversity in medical students, as previous career management research suggested that job values might, for instance, be dependent on social origins and gender. Cross-cultural research would also be beneficial to get an idea of the sociocultural dependence of these orientations, and the validity of the findings outside the context of this study.

Our findings have potential implications for career guidance and career choice skills training in medical students. Practically, examining and understanding career orientations among students can help in career counselling of medical students. Career management literature claims that self-awareness of values, interests and preferences in work and private life is essential in making appropriate career decisions [36]. Individuals' occupational decisions tend to be more appropriate and long-term satisfying when they are preceded by extensive career exploration, including self-awareness reflection [36]. In addition, goals are likely to be more realistic when they are based on accurate pictures of oneself and the career field.

On top of that stimulating students to start with exploring and prioritising their work values, preferences and motives might enable more appropriate and long-term satisfying decisions, and might be a valuable starting point to encourage students to consider a broader scope of medical specialties. Career choice theories all seem to include some form of cognitive reflection on motivation before a wide exploration. Although preferences and priorities might shift during clinical experiences, if students are offered the tools to reflect on, explore and better steer their own career choice process, they are able to use these skills and tools later on in their careers as well. This can prime students to make more rational decisions that are compatible with their values, interests and motives, with the potential of decreasing dropout during or after training [37]. The statement sorting exercise presented here actually might be an effective tool to help students to self-explore. Most students participating in this study found it useful for their own preference formation process, as they were challenged to think about the importance they attach to a variety of aspects relative to each other.

Before being able to intervene with career coaching, medical educationalists need to attend to the following conditions. First of all, being able to self-explore and learn the career management competencies in an effective way requires a safe non-judgmental environment, which might need some work in the setting of medical education. The safety might be restricted by messages the medical school or teachers consciously or unconsciously convey with regard to career opportunities in the medical field. Implicit beliefs and norms are transmitted to medical students in the so-called hidden curriculum, as a side effect of formal education [38]. As medical students try to adapt to the values of their teachers and role models, this hidden curriculum

may have a strong influence on medical students' willingness to openly discuss their values, needs and motives, especially when feeling that these deviate from their teachers or role models. Some students may even rethink their career aspirations when confronted with denigrating remarks about physicians in that particular specialty.

A second concern in the context of the mismatch between career aspirations of medical students and the needs for medical specialists in society is that the focus on the requirements for getting into medical school and postgraduate medical education might actually discourage any attempts for effective career management by students. The time invested in self-exploration and exploring potentially matching specialties cannot be invested in building a résumé that will help them get into medical specialty training, and can be perceived as a disadvantage to peers competing for the same training opportunities.

## Conclusion

We distinguished three career orientations among medical students in the Netherlands that differ in the importance of challenge, work-life balance, and recognition in relation to their career choices.

We argued that offering all students career coaching in which they are required to explore and prioritise their values, needs, and motivations is a good start. It might be promising to challenge students to broadly explore the medical specialties on their match with their identified values, needs, and motivations in order to actualise a better distribution over the specialties, while not risking dissatisfying careers. In any case, a better understanding of the career orientations of students is helpful in reconfiguring opportunities in the medical workforce in a way that helps achieve a better distribution over the specialties from a societal perspective, while still stimulating students to pursue a satisfying and sustainable medical career.

## Supporting information

**S1 Appendix. Questionnaire.**
(DOCX)

## Author Contributions

**Conceptualization:** Lokke Gennissen, Karen Stegers-Jager, Job van Exel, Lia Fluit, Jacqueline de Graaf, Matthijs de Hoog.

**Data curation:** Lokke Gennissen.

**Formal analysis:** Lokke Gennissen, Job van Exel.

**Methodology:** Lokke Gennissen, Karen Stegers-Jager, Job van Exel, Lia Fluit, Jacqueline de Graaf, Matthijs de Hoog.

**Project administration:** Lokke Gennissen.

**Supervision:** Karen Stegers-Jager, Job van Exel, Lia Fluit, Jacqueline de Graaf, Matthijs de Hoog.

**Writing – original draft:** Lokke Gennissen.

**Writing – review & editing:** Lokke Gennissen, Karen Stegers-Jager, Job van Exel, Lia Fluit, Jacqueline de Graaf, Matthijs de Hoog.

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
