## [Decision Letter · Decision Letter 0]

2 Nov 2020

PONE-D-20-29474

Career orientations of medical students: a Q-methodology study

PLOS ONE

Dear Dr. Gennissen,

Thank you for submitting your manuscript to PLOS ONE. After careful consideration, we feel that it has merit but does not fully meet PLOS ONE’s publication criteria as it currently stands. Therefore, we invite you to submit a revised version of the manuscript that addresses the points raised during the review process.

You have done a great job investigating career orientations of medical students using Q-methodology. However and per reviewers' recommendations, your manuscript requires minor revisions. Please consider revising your manuscript addressing those recommendations, and resubmit for consideration again.

We look forward to receiving your revised manuscript.

Kind regards,

Sina Safayi, D.V.M., Ph.D.

Academic Editor

PLOS ONE

Journal Requirements:

2. The data presented in PLOS ONE manuscripts must support the conclusions drawn (http://journals.plos.org/plosone/s/criteria-for-publication#loc-4).

Please ensure that the limitations of your study are sufficiently acknowledged and that your conclusions are directly supported by the data; in particular, please ensure that you provide a justification for drawing conclusions beyond the sample surveyed.

3.Thank you for including your ethics statement:  "The ethical review board of the Netherlands Association for Medical Education (Nederlandse Vereniging voor Medisch Onderwijs [NVMO]) approved the study (NERB number 812).".   

Please provide additional details regarding participant consent. In the ethics statement in the Methods and online submission information, please ensure that you have specified what type you obtained (for instance, written or verbal, and if verbal, how it was documented and witnessed). If your study included minors, state whether you obtained consent from parents or guardians. If the need for consent was waived by the ethics committee, please include this information.

4. We would like to ask the Corresponding Author to please provide their institutional email address.

5.We note that you have indicated that data from this study are available upon request. PLOS only allows data to be available upon request if there are legal or ethical restrictions on sharing data publicly. For information on unacceptable data access restrictions, please see http://journals.plos.org/plosone/s/data-availability#loc-unacceptable-data-access-restrictions.

Reviewers' comments:

Reviewer's Responses to Questions

**Comments to the Author**

1. Is the manuscript technically sound, and do the data support the conclusions?

Reviewer #1: Partly

Reviewer #2: Yes

Reviewer #3: Yes

Reviewer #4: Yes

Reviewer #5: Partly

Reviewer #6: Yes

2. Has the statistical analysis been performed appropriately and rigorously? 

Reviewer #1: Yes

Reviewer #2: Yes

Reviewer #3: Yes

Reviewer #4: I Don't Know

Reviewer #5: Yes

Reviewer #6: I Don't Know

3. Have the authors made all data underlying the findings in their manuscript fully available?

Reviewer #1: No

Reviewer #2: Yes

Reviewer #3: Yes

Reviewer #4: Yes

Reviewer #5: No

Reviewer #6: Yes

4. Is the manuscript presented in an intelligible fashion and written in standard English?

Reviewer #1: Yes

Reviewer #2: Yes

Reviewer #3: No

Reviewer #4: Yes

Reviewer #5: No

Reviewer #6: Yes

5. Review Comments to the Author

Reviewer #1: Thanks for the really interesting paper. It was terrific to read about the influence of student values on their career concepts; as this something that is rarely contemplated in formal higher education and medical training.

Your explanation of how you used Q methodology to analyse your problem was well done, particularly for a lay audience.

In your discussion on the selection of participants on page 6, I think it would have been more compelling and more rigorous had you expanded on the different “background characteristics” of the participants. It could be suggested that demographics, particularly cultural and socio-economic influencers would have an impact on the responses to the questions you asked the participants, more so than their gender and stage of study.

The collection of statements used and the analysis seemed appropriate for the research problem.

On pages 10 and 11 the phrase “defined by sorts of..” is used. This is not a typically utilized phrase in English and is confusing to the reader. I suggest a clearer statement is used to help the reader understand the participants who were identified.

It would have been more helpful to view the data in a graphical presentation so that the reader can quickly see the trends (a table is quite difficult to interpret and the values placed next the text content are hard to compare throughout the document). Perhaps a bar chart with positive and negative axes could be used.

Your discussion was interesting but requires further consideration and perhaps the addition of further data. In your conclusions (p14) you state that “our results offer an in-depth understanding of specialty preferences among medical students”. While I think you have certainly elucidated a commonality between med student values and the aspects of work that they find attractive (“career orientations”); you have yet to demonstrate a clear linkage between these aspects and specialities. It would be favourable to link the 3 identified career orientations with specific specialities so that you can then make this claim. Data from a focus group of practicing specialists may be required to where you could survey them for alignment of these to each speciality (For example what specialties most align to “achievement and recognition”).

Once again, a valuable investigation for the career development and guidance of medical students, but it may require some minor changes as discussed.

Reviewer #2: I appreciate the amount of work the authors have undertaken in this endeavor and have a few suggestions and clarifications that I hope may add to the value of this submitted manuscript.

1. in the introduction part, the authors forgot to give the introduction heading and started with the mismatch, which is really confusing and looks out of place.

2. In the method part, authors mentioned the JvE, KSJ and LG who revised the 62 statements. These abbreviations were not mentioned in the orther of authors. Please mention to avoid any misunderstanding.

Reviewer #3: Peer-review - Career orientations of medical students: A Q-methodology study

Summary of research and overall impression

The authors set out to understand the actual decision-making process medical students go through when choosing a specialty. The research aim and analytical approach are clearly stated as presented by the question of understanding the process of medical specialty preference formation and the utilization of Q-methodology. This hybrid methodology included qualitative and quantitative methods ensuring results could be presented in data form as well as presenting anonymized excerpts of participant voices to further support the quantitative findings. This study contributes to and advances the existing literature on specialty choice of medical students. This is a relevant and on-going topic that needs continued investigation and exploration.

The following is a breakdown of various sections of the study with compliments and comments related to clarification and technical structure.

Mismatch

The mismatch of supply of medical professionals and societal healthcare demands was approached by understanding the career aspirations of medical students. Specifically, addressing the actual decision-making process students go through when choosing a specialty. There is a grammatical error in line 41.

Career aspirations

This section presents a good vetting of the literature related to the understanding of medical specialty choice and does a good job of honing in on the decision-making process as the focal point for this study. There are grammar and punctuation errors in lines 81, 83 and 91.

Research aim

The research aim and analytical approach are clearly stated as presented by the question of understanding the process of medical specialty preference formation and the utilization of Q-methodology. There are grammar and punctuation errors in lines 102 and 103.

Methods

Study design, Context and Participants

The study design, context and participants sections are each well-written. There are grammar and punctuation errors in lines 120, 121.

Statement set

The process by which the Statement set was developed is thorough and seemingly well-vetted. The authors are serious and invested in the design of the Q-sort as is evidenced by their meticulous vetting process of reviewing the literature related to specialty preference, engaging clinical and medical education professionals in the design of the statement set, piloting the statement set among students, and revising the statements with feedback obtained to initiate the official study. There are grammar and punctuation errors in lines 146 and 149.

Data Collection

The description of the sorting process needs clarity and more detail in the description. The authors may want to consider using a series of visuals to depict each phase of the sorting process along with a more detailed description of each phase of the sorting process that ultimately leads to the final product in Figure 1. Since data is vital to any study, it is important to also include, perhaps as an addendum a list of the demographic questions on the paper-based questionnaire. Including these details will show transparency and enhance the validity of your overall study.

Analysis

The analysis, as it is framed, supports the study. For clarity on the idealized rankings, it would be helpful to include in the analysis, the statements that support each pattern.

Ethical Considerations

This section is well-outlined.

Results

The introductory paragraph of this section initiates a good summary and would be more strongly supported by explicitly stating the three main findings: life-long learning, work-life balance, and career achievement respectively. I suggest considering modification of the construct of the introductory paragraph to include these as it frames the detailed explanation you provide for each of the findings throughout the remainder of this section.

For each of the career orientations you have identified, you attempt to address their validity with your hybrid q-data. The information you present is supportive and might better serve the reader with consistent organization and more definitive grouping of the results as important, unimportant and neutral. I share the following for career orientation 1 as an example and follow with a note for career orientation 2:

Career orientation 1: Lifelong learning as a calling

It is helpful that you unpack the rankings and validity of the statements on your Q-sort that support this career orientation. As you reference Table 1, you should indicate which pattern (1, 2 or 3) to which this section relates. I question if you are attempting to take the results of this portion of the Q-sort and further organize the results into most important, neutral and unimportant? If so, it is best to explicitly state that for organizational purposes and understanding for readers.

Further, if I am following Table 1 correctly, the three highest ranked items supporting life-long learning are: Handling work mentally, opportunities for lifelong learning, work that can be handled physically and opportunities to get involved in research. Some of the data in Table 1 seems contrary to your explanation (for example, see statement 6). I could be misunderstanding your interpretation and if so, is there a way to construct your explanation to be clearer about the intention of those students who value lifelong learning as a calling?

Career orientation 2: Work-life balance

For consistency, begin this section the same way you began the last section – use the same structure of phrasing in your opening sentence as the previous section.

Discussion

The discussion section is well-organized; synthesizes the career orientation patterns found in the study; and supports salient and on-going issues broadly faced in the medical profession: attrition, work-life balance, generational differences among others. The authors also appropriately questioned the use of Q-sorting as a career guidance tool and other possible career counseling or career coaching activities. The authors identified appropriate study limitations as well as recommendations for future research.

Of particular note in the discussion section is the suggestion of intentional career development activities at the on-set as well as throughout medical school training to provide students with the opportunity for an informed perspective when it comes to making a specialty choice.

There are grammatical and punctuation errors on lines 315, 316, 372, 374

Reviewer #4: Thank you for the opportunity to review this manuscript. As noted above, my recommendation is for this paper to be published with minor - I hope - revisions. The overall narrative of the paper is clear, however, I note 7 ways in which I think this manuscript can be improved:

1. The body of the paper is does not have major issues, but the introduction starts with a frame of reference on the lack of adequate numbers of medical professionals in certain specialties then transitions to a separate problem of the connection between student interests and the alignment with their career selection. I think the strength of this paper would be improved with a more explicit link between what the authors see as the connection between career exploration and the lack of enough medical professionals in certain specialities. I think this could be addressed by adding, close to the end of the paper, their perspective on the relationship between these two topics.

2. The authors raise an important point in the middle of the discussion - regarding the homogeneity (or possible homogeneity - of the students who are admitted to medical school. If the data reflect what is a problem with diversity of the student body and reflects the criteria for admission, my instinct is that perhaps one way to consider how to ensure who can fill these various speciality gaps would be to consider who gets into medical school.

3. Line 118...The authors write "Demographic characteristics of their students and their curricula are quite similar." How do they know that: can they provide contextual data to prove/clarify that? Which characteristics of the students are they thinking about? Do they mean gender alone? Or are they consider socioeconomic. geographic, other background factors that could be relevant? In line 335, the authors comments on the homogeneity of the population and so this information could be important.

4. The discussion of the paper might be enriched by a commentary on the "year of progress" in which students landed, particularly for Career Orientations 1 and 2. In reading, I noted that Cohort 1 ("life long learning") were years 1-4 students while Cohort 2 ("work-life balance") were years 2-6. While the qualitative approaches in this study accurately address their questions, and I do not want to encourage overstating results from a limited number of participants, I think this observation is noteworthy: does more experience in the field turn the optimism of lifelong learning to work-life balance, for example, correlate with life events?

5. Line 155...I am not sure that I understand what "labour content" means, particularly considering the parallel structure in which it is included alongside other terms.

6. At line 220, I was first prompted to consider the impact of the ordering of the terms in the list study participants were asked to rank. How was the order determined? Table 1 suggests it was not presented as an alphabetical list. Is there any reason to be concerned about the influence of the order of the list and the way people responded? If so, this should be addressed in the results/discussion.

7. Finally, the overall writing is good. I noted some grammatical/punctuation errors which do not change the context of the statements, but could be corrected to further strengthen the manuscript:

...lines 14/15...."many countries are struggling with a misalignment of specialty preferences of students and (add) THEIR societal needs regarding (add) THE future medical workforce."

...line 38...."health professionals to societal demands is essential in aspiring (add)TO CREATE an affordable health"

...line 138..."agreed (add)TO participation, completed the study. "

...line 372..."there are conditions which (change) <<has to="">> HAVE to be met,</has>

Reviewer #5: Dear Authors,

Thank you for undertaking this study and writing up your findings. The question of trainee career path selection is important and timely and I commend you for describing the importance (and sometimes confounding) effect of faculty bias and university culture on medical student career choice.

I have some comments and concerns related to study design which lead me to answer "partly" for question #1.

The stated research aim is "to contribute to the understanding of the process of medical specialty preference formation, by focusing on student career orientations." (line 101) The Q methodology employed sheds light on student work place preferences, values and motivations, but there was no mention of the students' career path preferences. It would be a more valuable study if study participants who were grouped into these three categories were also asked which medical specialties they were highly considering. If the subgroups are each interested in the same specialty fields that would go a long way to explaining the mismatch that is described in some detail in the introduction.

24 study participants seems like a low number. Missing from the methods section is how many total students are in these two medical training programs. For example do these 24 study participants equate to 10% of the students or 90%. Why did the authors not invite all students to participate? Perhaps the answer is that there was not enough staff time to conduct the 1-1 interviews.

Relatedly, line 198 describes the gender breakdown and other demographics of the study participants. How does this relate to the demographics of the entire student population. The answer to this question is important in predicting if there was selection bias in the study. If the demographics of the study participants and complete student populations are different then that should be discussed in the limitations section.

More explanation or a citation is needed to describe what is meant on line 173 by varimax rotation analysis.

The abstract states (line 30) "These insights can help reconfigure opportunities in the medical workforce and shift career choices of medical students closer towards future needs in society." I would say that this statement is an overreach given the study design and results. There is no discussion about the likely specialties chosen by the groupings identified in the study. Nor are there suggestions given in the paper for how to reconfigure opportunities in the medical workforce, based on the findings.

The strengths of the study include 1) reconfirming the importance of career counseling for students in professional and graduate training; 2) acknowledging the discontent among rising medical professionals with some aspects of the current medical workplace; 3) encouraging early career guidance and facilitated introspection to help medical students prepare for a satisfying career.

In response to question #4 there are multiple typographical and grammatical errors in the document.

Reviewer #6: I found the article “Career orientations of medical students: a Q-methodology study” a worthwhile contribution to understanding the unconscious preferences that medical students may have in choosing their careers, and how further exploring these preferences may help guide medical students towards satisfying professions. The authors suggest that a deeper examination of medical students’ career goals may assist in the ability of societies to fill unmet needs in the profession, either directly by retaining physicians and preventing burnout, or as they indirectly imply, by encouraging the students to pursue a career that better matches the needs of a society.

The manuscript appears to be novel, has merit and is well written, yet there are some minor revisions suggested to help improve the manuscript.

1. One of the references they cited (Cleland JA et al.) used surveys of 810 students. Q-methodology understandably requires a lot more commitment from the human subjects, and therefore can be expected to have a smaller number of participants, yet it might be helpful to demonstrate that 24 students were a big enough sample size to justify their conclusions.

2. The authors seem to imply that self-examination of long-term priorities and goals of medical students will help prevent burnout (understood), yet they also seem to indirectly imply that this practice will help fulfill societal needs that are currently unmet. This concept was brought up at the beginning (lines 29-47) and end of the paper (lines 400-403). For example, in lines 29-32, they state “These insights can help reconfigure opportunities in the medical workforce and shift career choices of medical students closer towards future needs in society.” Do they seem to imply that increasing the awareness of medical students who value work-life balance will increase the number of medical students pursuing a career in elderly care or public health (societal needs stated in the manuscript)? Please elaborate.

3. The authors compare how the Q-test differs from similar assessments of medical student career orientation (lines 63-70). Is this Q-test assessment of medical student career orientation to your knowledge novel? If so, please state so. Can your results be compared with Q-test analysis of career choices in other health care fields (pharmacists, dentists, nurses etc.)?, it seems like there was at least one in my brief search that discussed career choices of pharmacy students. Did other studies show potential in the self-analysis process helping meet societal needs for occupations in health care delivery?

4. Much of the statistical jargon used might not be readily understood by the casual reader of PLOS ONE. Perhaps a brief explanation would help the casual reader for some of the topics. For example,

“sampling and snowballing”, “convenience sample” in lines 130-131.

“JvE, KSJ and LG” on line 153

“By-person factor analysis, centroid factor analysis, varimax rotation” line 172-173- is it possible to provide a brief 1-2 sentence explanation describing how these showed a maximum of three factors?

5. Is it possible to include an example of the paper-based questionnaire (without student answers)? Line 170

6. Please clarify the use of “participants to identify consecutive participants” (line 134), does this mean that students participating in the study would recruit other students to participate in the study? Would this potentially hurt the diversity of students in the study? Could this potentially explain why 15 out of the 24 subjects were female?

7. Approximately how many clinical and medical education professionals participated in the creation of the statement set (lines 143-144)?

8. Is it possible to include a figure describing how a maximum of 3 career orientations could explain 48% of the variance. This is a major part of the study and it would be helpful to help the novice reader understand how the data can be used to show that three different career paths could be obtained from the data.

9. Do the authors have any comment/discussion on how work-life balance was the one career path overwhelmingly chosen by female students?

10. Some of the discussion can be consolidated and shortened e.g. “Medical students with a work-life balance orientation to their career primarily expressed a desire for a good balance between work and private life” lines 284-286

11. Was there a survey to demonstrate that the students felt that the practice benefited their values, needs and motivations? Authors indicated the students felt that they benefited from the exercise, yet if no official survey was done, they should state so. Lines 315-317, and 367-368

12. Table 1 – please label Pattern 1, 2, 3 with regards to the indicated career orientation, e.g. work-life balance

13. Is it possible to include additional tables that rank each of the statement for the three career orientations?

For example Career orientation 1: List most important at top, least important at bottom, do for the other career orientations. Table 1 by itself is hard to quickly process by itself.

6. PLOS authors have the option to publish the peer review history of their article (what does this mean?). If published, this will include your full peer review and any attached files.

Reviewer #1: No

Reviewer #2: No

Reviewer #3: No

Reviewer #4: No

Reviewer #5: No

Reviewer #6: No

---

## [Author Response · Author response to Decision Letter 0]

15 Feb 2021

We would like to thank the editor and the reviewers for their thorough review. 

General comments 

We carefully revised the manuscript following these requirements. 

2. The data presented in PLOS ONE manuscripts must support the conclusions drawn (http://journals.plos.org/plosone/s/criteria-for-publication#loc-4).

Please ensure that the limitations of your study are sufficiently acknowledged and that your conclusions are directly supported by the data; in particular, please ensure that you provide a justification for drawing conclusions beyond the sample surveyed.

We ensured that the limitations of the study are sufficiently acknowledged, that conclusions are supported by the data and we were more explicit in conclusions beyond the sample surveyed.

3.Thank you for including your ethics statement: "The ethical review board of the Netherlands Association for Medical Education (Nederlandse Vereniging voor Medisch Onderwijs [NVMO]) approved the study (NERB number 812).". 

Please provide additional details regarding participant consent. In the ethics statement in the Methods and online submission information, please ensure that you have specified what type you obtained (for instance, written or verbal, and if verbal, how it was documented and witnessed). If your study included minors, state whether you obtained consent from parents or guardians. If the need for consent was waived by the ethics committee, please include this information.

All participants gave written informed consent. (page 10 line 219, 220 of the manuscript without track changes and page 11 line 259-560 in the manuscript with track changes)

4. We would like to ask the Corresponding Author to please provide their institutional email address.

The corresponding author provided her institutional email address.

5.We note that you have indicated that data from this study are available upon request. PLOS only allows data to be available upon request if there are legal or ethical restrictions on sharing data publicly. For information on unacceptable data access restrictions, please see http://journals.plos.org/plosone/s/data-availability#loc-unacceptable-data-access-restrictions.

The reason for having data available on request is the small sample size, causing ethical restrictions when providing both the demographic characteristics of the participants in combination with the Q-sort data. Data could traceable to the individual. 

Comments to the Author

Reviewer #1: 

Thanks for the really interesting paper. It was terrific to read about the influence of student values on their career concepts; as this something that is rarely contemplated in formal higher education and medical training.

Your explanation of how you used Q methodology to analyse your problem was well done, particularly for a lay audience.

We thank the reviewer for investing time in reviewing our manuscript and for his/her kind words.

In your discussion on the selection of participants on page 6, I think it would have been more compelling and more rigorous had you expanded on the different “background characteristics” of the participants. It could be suggested that demographics, particularly cultural and socio-economic influencers would have an impact on the responses to the questions you asked the participants, more so than their gender and stage of study.

We agree with the reviewer that the cultural and social background characteristics of the participants could have impact on the rankings of the participants. Therefore, we added a table with the background characteristics of the participants on page 25, 26 of the revised manuscript without track changes and on page 27-28 of the revised manuscript with the track changes. 

The collection of statements used and the analysis seemed appropriate for the research problem.

On pages 10 and 11 the phrase “defined by sorts of..” is used. This is not a typically utilized phrase in English and is confusing to the reader. I suggest a clearer statement is used to help the reader understand the participants who were identified.

We replaced “defined by sorts of ..” by “was represented by the Q-sorts of..” on page 11, line 249 and page 12 line 269 and page 13 line 290 of the revised manuscript without track changes and on page 13, line 294 and line 314 and on page 14 line 335 of the revised manuscript with the track changes. 

It would have been more helpful to view the data in a graphical presentation so that the reader can quickly see the trends (a table is quite difficult to interpret and the values placed next the text content are hard to compare throughout the document). Perhaps a bar chart with positive and negative axes could be used.

By adding table 3, 4 and 5 (tables with characterising statement per career orientation) on page 29, 30 and 31 of the manuscript without track changes we aimed to make the data easier to oversee and retrieve. 

Additionally, we removed information which is less relevant from table 2 to make it easier to oversee the relevant data. 

Your discussion was interesting but requires further consideration and perhaps the addition of further data. In your conclusions (p14) you state that “our results offer an in-depth understanding of specialty preferences among medical students”. While I think you have certainly elucidated a commonality between med student values and the aspects of work that they find attractive (“career orientations”); you have yet to demonstrate a clear linkage between these aspects and specialities. It would be favourable to link the 3 identified career orientations with specific specialities so that you can then make this claim. Data from a focus group of practicing specialists may be required to where you could survey them for alignment of these to each speciality (For example what specialties most align to “achievement and recognition”).

Initially, linking career orientations to medical specialties was also our line of thinking. However, while analysing these Q-sorts, we started to realize that the professional content statements were not the distinguishing factors between these career orientations. Within specialties a medical specialist can have different career orientations. Therefore matching career orientations to specific specialties is not helpful based on our data. This made us realize that probably the gain of aligning the medical students preferences and future societal needs probably is in a broad exploration based on their priorities which become apparent in their Q-sorts. In that exploration they can become aware that there are more suitable career options than they may have expected in the first place.

We elaborated on these considerations in the discussion:

“It is tempting to link a specific career orientations to certain medical specialties. Although this would certainly be interesting to further explore in future research, there is a pitfall to it. First of all, this study showed us the work-related content statements were not the distinguishing factors between these career orientations. That raises the question whether different career orientations can be found in medical specialists in a medical specialty. Secondly, linking certain career orientations to medical specialties might inhibit the exploration phase, risking reinforcement of the maldistribution over the medical specialties.” Page 16, line 379-385 of the revised manuscript without track changes and on page 18, line 436-442 of the revised manuscript with the track changes. 

Once again, a valuable investigation for the career development and guidance of medical students, but it may require some minor changes as discussed.

Thank you. We feel our manuscript improved by your suggestions. 

Reviewer #2: 

I appreciate the amount of work the authors have undertaken in this endeavor and have a few suggestions and clarifications that I hope may add to the value of this submitted manuscript.

We thank the reviewer for his/her review and suggestions to improve our manuscript. 

1. in the introduction part, the authors forgot to give the introduction heading and started with the mismatch, which is really confusing and looks out of place.

We added the heading Introduction (on page 3 line 36 of the revised manuscript without track changes and on page 3 line 37 of the revised manuscript with the track changes). 

2. In the method part, authors mentioned the JvE, KSJ and LG who revised the 62 statements. These abbreviations were not mentioned in the order of authors. Please mention to avoid any misunderstanding.

We added the abbreviations of the authors in the order of the authors (on page 1 line 2,3 of the revised manuscript without track changes and on page 1, line 2,3 of the revised manuscript with the track changes). 

Reviewer #3: 

Summary of research and overall impression

The authors set out to understand the actual decision-making process medical students go through when choosing a specialty. The research aim and analytical approach are clearly stated as presented by the question of understanding the process of medical specialty preference formation and the utilization of Q-methodology. This hybrid methodology included qualitative and quantitative methods ensuring results could be presented in data form as well as presenting anonymized excerpts of participant voices to further support the quantitative findings. This study contributes to and advances the existing literature on specialty choice of medical students. This is a relevant and on-going topic that needs continued investigation and exploration.

We thank the reviewer for his/her thorough reviewing of our manuscript and for his/her kind words. 

The following is a breakdown of various sections of the study with compliments and comments related to clarification and technical structure.

Mismatch

The mismatch of supply of medical professionals and societal healthcare demands was approached by understanding the career aspirations of medical students. Specifically, addressing the actual decision-making process students go through when choosing a specialty. There is a grammatical error in line 41.

We rephrased line 41 to correct the grammatical error. 

Career aspirations

This section presents a good vetting of the literature related to the understanding of medical specialty choice and does a good job of honing in on the decision-making process as the focal point for this study. There are grammar and punctuation errors in lines 81, 83 and 91.

The grammar and punctuation errors in lines 81, 83 and 91 were revised. 

Research aim

The research aim and analytical approach are clearly stated as presented by the question of understanding the process of medical specialty preference formation and the utilization of Q-methodology. There are grammar and punctuation errors in lines 102 and 103.

The grammar and punctuation errors in lines 102 and 103 were revised. 

Methods

Study design, Context and Participants

The study design, context and participants sections are each well-written. There are grammar and punctuation errors in lines 120, 121.

The grammar and punctuation errors in lines 120, 121 were revised. 

Statement set

The process by which the Statement set was developed is thorough and seemingly well-vetted. The authors are serious and invested in the design of the Q-sort as is evidenced by their meticulous vetting process of reviewing the literature related to specialty preference, engaging clinical and medical education professionals in the design of the statement set, piloting the statement set among students, and revising the statements with feedback obtained to initiate the official study. There are grammar and punctuation errors in lines 146 and 149.

The grammar and punctuation errors in lines 146, 149 were revised. 

Data Collection

The description of the sorting process needs clarity and more detail in the description. The authors may want to consider using a series of visuals to depict each phase of the sorting process along with a more detailed description of each phase of the sorting process that ultimately leads to the final product in Figure 1. Since data is vital to any study, it is important to also include, perhaps as an addendum a list of the demographic questions on the paper-based questionnaire. Including these details will show transparency and enhance the validity of your overall study.

We rephrased the description of the sorting process with the aim of a more clear description of the process (on page 8 line 176-190 of the revised manuscript without track changes and on page 8, 9 line 189-205 of the revised manuscript with the track changes). The suggestion of using visuals is very helpful, we added figures explaining the sorting process (on page 25 of the revised manuscript without track changes and on page 26-27 of the revised manuscript with the track changes). 

We added the list of the demographic questions on the paper-based questionnaire to the appendix (on page 32 of the revised manuscript without track changes and on page 34. of the revised manuscript with the track changes). 

Analysis

The analysis, as it is framed, supports the study. For clarity on the idealized rankings, it would be helpful to include in the analysis, the statements that support each pattern.

The idealized rankings are the result of the analysis. Based on the consensus and distinguishing statements one could interpret the factors resulting in the description of the career orientations. As Q-sort is a mixed method, one could not claim that the distinguishing factors are the only factors that support the pattern. To give more insight on the starting point for the interpretation we added tables with the characterising statements of the career orientations. These tables can be found on page 29, 30, and 31 of the revised manuscript without track changes and on page 31, 32, and 33 of the revised manuscript with the track changes. 

Ethical Considerations

This section is well-outlined.

Results

The introductory paragraph of this section initiates a good summary and would be more strongly supported by explicitly stating the three main findings: life-long learning, work-life balance, and career achievement respectively. I suggest considering modification of the construct of the introductory paragraph to include these as it frames the detailed explanation you provide for each of the findings throughout the remainder of this section.

We followed your suggestion and the results section is now more clear by explicitly stating the three main findings in the introductory paragraph: “These career orientations will be referred to as life-long learning as a calling, work-life balance and achievement and recognition.” These changes can be found on page 10 line 235, 236, 237 of the revised manuscript without track changes and on page 12, line 278, 279 of the revised manuscript with the track changes.

For each of the career orientations you have identified, you attempt to address their validity with your hybrid q-data. The information you present is supportive and might better serve the reader with consistent organization and more definitive grouping of the results as important, unimportant and neutral. I share the following for career orientation 1 as an example and follow with a note for career orientation 2:

We understand the suggestion for a more consistent organization, however, as often within q-sort methodology there are no thresholds to precisely determine which statements are important, neutral or unimportant. We interpreted and described the orientations by determining statements’ relative importance, i.e. being more important or less important than other statements within a particular orientation. Since it was the relative importance we were most interested in, we did not ask our participants to define the borders between important, neutral and unimportant and we let them at any time of the ordering move statements if they felt this was necessary to display their viewpoint best. So, if we would organize the statements in important, neutral and unimportant this would be our grouping of the statements, not that of the participants. Furthermore, the interpretation also includes the qualitative data. All in all, we therefore feel a consistent organization is not feasible. 

Career orientation 1: Lifelong learning as a calling

It is helpful that you unpack the rankings and validity of the statements on your Q-sort that support this career orientation. As you reference Table 1, you should indicate which pattern (1, 2 or 3) to which this section relates. I question if you are attempting to take the results of this portion of the Q-sort and further organize the results into most important, neutral and unimportant? If so, it is best to explicitly state that for organizational purposes and understanding for readers.

We added references with orientation names to the table with the idealized rankings (page 27 and 28 of the manuscript without track changes, page 29-30 of the manuscript with track changes). As stated in reaction to the previous comment we do not feel it is feasible to further organize the results in most important, neutral and unimportant, since we do not have thresholds to classify statements in one of the three categories. It is the relative importance to other statements which distinguishes the career orientations and therefore is described in the results. 

Further, if I am following Table 1 correctly, the three highest ranked items supporting life-long learning are: Handling work mentally, opportunities for lifelong learning, work that can be handled physically and opportunities to get involved in research. Some of the data in Table 1 seems contrary to your explanation (for example, see statement 6). I could be misunderstanding your interpretation and if so, is there a way to construct your explanation to be clearer about the intention of those students who value lifelong learning as a calling?

Although consensus, distinguishing factors and the statements at the extreme ends take an important role in coming to an interpretation, it is the interrelationship of the many items within the career orientation that should drive our interpretation of the career orientation. One searches for patterns in the career orientations and relates these possible patterns with the qualitative data from the interviews. This means that the emphasis might be more on patterns found in the midsection than on the extreme ends and that there might be an emphasis on statements which are not distinguishing statements. We rephrased the analysis part of the methods. We added tables with the characterising statements of each career orientation, with the aim to make it easier to find these results. These changes can be found on page 29, 30, and 31 of the revised manuscript without track changes and on page 31, 32, and 33 of the revised manuscript with the track changes

Career orientation 2: Work-life balance

For consistency, begin this section the same way you began the last section – use the same structure of phrasing in your opening sentence as the previous section.

We changed the phrase about the demographic characteristics of the represented participants in the orientations to be congruent. These changes can be found on page 11, line 249, 250 of the revised manuscript without track changes and on page 13 line 295 of the revised manuscript with the track changes.

Discussion

The discussion section is well-organized; synthesizes the career orientation patterns found in the study; and supports salient and on-going issues broadly faced in the medical profession: attrition, work-life balance, generational differences among others. The authors also appropriately questioned the use of Q-sorting as a career guidance tool and other possible career counseling or career coaching activities. The authors identified appropriate study limitations as well as recommendations for future research.

Of particular note in the discussion section is the suggestion of intentional career development activities at the on-set as well as throughout medical school training to provide students with the opportunity for an informed perspective when it comes to making a specialty choice.

There are grammatical and punctuation errors on lines 315, 316, 372, 374

Grammatical and punctuation errors in lines 315, 316, 372 and 374 were revised.

Reviewer #4: 

Thank you for the opportunity to review this manuscript. As noted above, my recommendation is for this paper to be published with minor - I hope - revisions. The overall narrative of the paper is clear, however, I note 7 ways in which I think this manuscript can be improved:

Thank you for constructive review of our manuscript. 

1. The body of the paper is does not have major issues, but the introduction starts with a frame of reference on the lack of adequate numbers of medical professionals in certain specialties then transitions to a separate problem of the connection between student interests and the alignment with their career selection. I think the strength of this paper would be improved with a more explicit link between what the authors see as the connection between career exploration and the lack of enough medical professionals in certain specialities. I think this could be addressed by adding, close to the end of the paper, their perspective on the relationship between these two topics.

We feel we strengthened our manuscript by making a more explicit link between the career exploration and the lack of enough medical professionals in certain specialties. Below the parts of the manuscript we revised to make this link more clear: 

“This also creates opportunities to address the maldistribution by on one hand opening a new window on medical specialties to become more attractive, while on the other hand guiding students to the insight what features of future work are most important to them and stimulating them to explore the medical specialties accordingly. Many features (e.g. “working in a multidisciplinary team”) can be found in multiple specialties. Showing this to students by enabling a broad exploration might be a promising start in striving towards a better distribution over the specialties.” (page 16, line 362-368 of the revised manuscript without track changes and on page17, line 414-421 of the revised manuscript with the track changes).

“We argued that offering all students career coaching in which they are required to explore and prioritise their values, needs, and motivations is a good start. It might be promising to challenge students to broadly explore the medical specialties on their match with their identified values, needs, and motivations in order to actualise a better distribution over the specialties, while not risking dissatisfying careers.” 

(on page 19 line 452-459 of the revised manuscript without track changes and on page 21 line 510-517 of the revised manuscript with the track changes). 

2. The authors raise an important point in the middle of the discussion - regarding the homogeneity (or possible homogeneity - of the students who are admitted to medical school. If the data reflect what is a problem with diversity of the student body and reflects the criteria for admission, my instinct is that perhaps one way to consider how to ensure who can fill these various speciality gaps would be to consider who gets into medical school.

We agree with the reviewer on the possible relation with the selection for medical school. This is, in our view, important in multiple ways. First of all, the type of person one selects could have an influence on later specialty preferences. Secondly, the criteria for admission and the message we are sending out to potential medical students on what is important could have a major influence on specialty preferences. However the medical school selection is years apart from the medical specialty choice, in which there are probably multiple factors of influence on medical specialty preference. Therefore, we do not think we could solve the misalignment of student preferences for medical specialties and future societal needs by solely changing the selection for medical school. Based on these arguments, we decided not the further elaborate on the role of selection of medical school in the manuscript. 

3. Line 118...The authors write "Demographic characteristics of their students and their curricula are quite similar." How do they know that: can they provide contextual data to prove/clarify that? Which characteristics of the students are they thinking about? Do they mean gender alone? Or are they consider socioeconomic. geographic, other background factors that could be relevant? In line 335, the authors comments on the homogeneity of the population and so this information could be important.

We added a reference to support our statement of the similarity of the different medical schools: “Demographic characteristics of their students and their curricula are quite similar [24].” (on page 6, line 128-129 of the revised manuscript without track changes and on page 6, line 137-138 of the revised manuscript with the track changes).

4. The discussion of the paper might be enriched by a commentary on the "year of progress" in which students landed, particularly for Career Orientations 1 and 2. In reading, I noted that Cohort 1 ("life long learning") were years 1-4 students while Cohort 2 ("work-life balance") were years 2-6. While the qualitative approaches in this study accurately address their questions, and I do not want to encourage overstating results from a limited number of participants, I think this observation is noteworthy: does more experience in the field turn the optimism of lifelong learning to work-life balance, for example, correlate with life events?

This is indeed an interesting observation. We added an elaboration on these findings in the discussion section: “Students in the work-life balance also were further along their education. This prompts the question whether more experience in the field will push students to the work-life orientation or that life-events along the way will make students turn to this career orientation.” (page 15, line 339-341 of the revised manuscript without track changes and on page 16 line 388-392 of the revised manuscript with the track changes).

5. Line 155...I am not sure that I understand what "labour content" means, particularly considering the parallel structure in which it is included alongside other terms.

We changed the term “labour content” to “work-related content” (page 8, line 173 of the revised manuscript without track changes and on page 8, line 186 of the revised manuscript with the track changes). 

6. At line 220, I was first prompted to consider the impact of the ordering of the terms in the list study participants were asked to rank. How was the order determined? Table 1 suggests it was not presented as an alphabetical list. Is there any reason to be concerned about the influence of the order of the list and the way people responded? If so, this should be addressed in the results/discussion.

The statements were presented to the participants in a random order (see page 8, line 176, 177 of the revised manuscript without track changes and page 9 line 190, 191in the revised manuscript with track changes). 

7. Finally, the overall writing is good. I noted some grammatical/punctuation errors which do not change the context of the statements, but could be corrected to further strengthen the manuscript:

...lines 14/15...."many countries are struggling with a misalignment of specialty preferences of students and (add) THEIR societal needs regarding (add) THE future medical workforce."

...line 38...."health professionals to societal demands is essential in aspiring (add)TO CREATE an affordable health"

...line 138..."agreed (add)TO participation, completed the study. "

...line 372..."there are conditions which (change) <> HAVE to be met,

We have corrected these sentences. 

Reviewer #5: 

Dear Authors,

Thank you for undertaking this study and writing up your findings. The question of trainee career path selection is important and timely and I commend you for describing the importance (and sometimes confounding) effect of faculty bias and university culture on medical student career choice.

Thank you for reviewing our manuscript. Thank you for the kind words. 

I have some comments and concerns related to study design which lead me to answer "partly" for question #1.

The stated research aim is "to contribute to the understanding of the process of medical specialty preference formation, by focusing on student career orientations." (line 101) The Q methodology employed sheds light on student work place preferences, values and motivations, but there was no mention of the students' career path preferences. It would be a more valuable study if study participants who were grouped into these three categories were also asked which medical specialties they were highly considering. If the subgroups are each interested in the same specialty fields that would go a long way to explaining the mismatch that is described in some detail in the introduction.

In our demographic questionnaire we asked students what their current medical specialty preference was. We could not find very obvious patterns in the distribution of medical specialty preferences in the different career orientations. 

However, our aim with this study was to find underlying values and motivations, not so much the actual specialty choice. This would be an interesting direction for future research, but we would not be surprised to see that medical specialists within a particular specialty could have different career orientations.

24 study participants seems like a low number. Missing from the methods section is how many total students are in these two medical training programs. For example do these 24 study participants equate to 10% of the students or 90%. Why did the authors not invite all students to participate? Perhaps the answer is that there was not enough staff time to conduct the 1-1 interviews.

We understand that the small number of participants raised questions. This number is not based on logistic restrictions. Because the used methodology is not strengthened by large sample sizes, we invested in a thorough development of the Q-set and in the sampling strategy. A limited number of well-selected participants is sufficient to reach saturation and to identify the main viewpoints about a topic.[23] 

Relatedly, line 198 describes the gender breakdown and other demographics of the study participants. How does this relate to the demographics of the entire student population. The answer to this question is important in predicting if there was selection bias in the study. If the demographics of the study participants and complete student populations are different then that should be discussed in the limitations section.

To give more insight in how our participants’ demographics relate to the demographics of the entire student population we added: “

These twenty-four students were from medical schools at two universities, 9 were male and 15 were female. The mean age of participants was 21.3, with a range of 18-27 years. This gender and age distribution is representative of the total student populations.[25] Five participants were in the first year of medical school, three of them in the second year, four in the third year, four in the fourth year, four in the fifth year and three in the sixth year.” (page 10 line 227-231 of the manuscript without track changes and page 12, line 267-273 of the manuscript with the track changes). 

More explanation or a citation is needed to describe what is meant on line 173 by varimax rotation analysis.

We rephrased the analysis part of the method section to be more clear on our analysis and we added a citation with regard to the varimax rotation analysis. 

“By-person factor analysis was used to identify distinct career orientations from the individual ranking data.[23] First, a correlation matrix between the rankings of the statements by the participants is computed. Assuming that if two participants rank the 62 statements in a similar way, they have a similar career orientation, factor analysis (i.e., centroid factor extraction followed by varimax rotation [23]) is then applied to identify the main patterns in the ranking data. The selection of the number of patterns (or factors) to extract from the data is based on statistical criteria (i.e., Eigenvalue >1 and a minimum of two participants statistically significantly associated with the pattern) and whether patterns have a coherent interpretation that is also supported by the corresponding qualitative data from the interviews.” (page 9 line 194-202 of the manuscript without track changes and page line 209-217 of the manuscript with the track changes). 

The abstract states (line 30) "These insights can help reconfigure opportunities in the medical workforce and shift career choices of medical students closer towards future needs in society." I would say that this statement is an overreach given the study design and results. There is no discussion about the likely specialties chosen by the groupings identified in the study. Nor are there suggestions given in the paper for how to reconfigure opportunities in the medical workforce, based on the findings.

We rephrased the sentence: “This knowledge can help to design interventions to shift career choices of medical students closer towards future needs in society.” page 1 line 30-31 of the manuscript without track changes and page 2 line 30, 31 of the manuscript with the track changes).

Besides rephrasing the sentence we added more explanation on the link between knowledge on career orientations and meeting future societal workforce needs (more details in reaction to reviewer 4, first comment). 

The strengths of the study include 1) reconfirming the importance of career counseling for students in professional and graduate training; 2) acknowledging the discontent among rising medical professionals with some aspects of the current medical workplace; 3) encouraging early career guidance and facilitated introspection to help medical students prepare for a satisfying career.

Thank you for your kind words. 

In response to question #4 there are multiple typographical and grammatical errors in the document.

We reviewed the manuscript on typographical and grammatical errors. 

Reviewer #6: 

I found the article “Career orientations of medical students: a Q-methodology study” a worthwhile contribution to understanding the unconscious preferences that medical students may have in choosing their careers, and how further exploring these preferences may help guide medical students towards satisfying professions. The authors suggest that a deeper examination of medical students’ career goals may assist in the ability of societies to fill unmet needs in the profession, either directly by retaining physicians and preventing burnout, or as they indirectly imply, by encouraging the students to pursue a career that better matches the needs of a society.

The manuscript appears to be novel, has merit and is well written, yet there are some minor revisions suggested to help improve the manuscript.

Thank you for investing time in our manuscript. Thanks for the kind words. We appreciate the suggestions to improve our manuscript. 

1. One of the references they cited (Cleland JA et al.) used surveys of 810 students. Q-methodology understandably requires a lot more commitment from the human subjects, and therefore can be expected to have a smaller number of participants, yet it might be helpful to demonstrate that 24 students were a big enough sample size to justify their conclusions.

As mentioned above in response to reviewer #5, the used methodology is not strengthened by large sample sizes. In Q-sort a limited number of well-selected participants is sufficient to reach saturation and to identify the main viewpoints about a topic. A thorough development of a Q-sort statement set and well-sampled participants does strengthen the methodology. 

2. The authors seem to imply that self-examination of long-term priorities and goals of medical students will help prevent burnout (understood), yet they also seem to indirectly imply that this practice will help fulfill societal needs that are currently unmet. This concept was brought up at the beginning (lines 29-47) and end of the paper (lines 400-403). For example, in lines 29-32, they state “These insights can help reconfigure opportunities in the medical workforce and shift career choices of medical students closer towards future needs in society.” Do they seem to imply that increasing the awareness of medical students who value work-life balance will increase the number of medical students pursuing a career in elderly care or public health (societal needs stated in the manuscript)? Please elaborate.

We elaborated that on one hand reconfiguring medical specialties and on the other hand guiding students to broadly explore on the basis of their priorities might be an interesting start for striving towards a better distribution. We added: “This also creates opportunities to address the maldistribution by on one hand opening a new window on medical specialties to become more attractive, while on the other hand guiding students to the insight what features of future work are most important to them and stimulating them to explore the medical specialties accordingly. Many features (e.g. “working in a multidisciplinary team”) can be found in multiple specialties. Showing this to students by enabling a broad exploration might be a promising start in striving towards a better distribution over the specialties.” (page 16 line 362-368 of the revised manuscript without track changes and on page 17, 18, line 414-421 of the revised manuscript with the track changes).

3. The authors compare how the Q-test differs from similar assessments of medical student career orientation (lines 63-70). Is this Q-test assessment of medical student career orientation to your knowledge novel? If so, please state so. Can your results be compared with Q-test analysis of career choices in other health care fields (pharmacists, dentists, nurses etc.)?, it seems like there was at least one in my brief search that discussed career choices of pharmacy students. Did other studies show potential in the self-analysis process helping meet societal needs for occupations in health care delivery?

We added: “Although Q-sort methodology has previously been used in career research, this study is the first to apply a Q-sort approach to medical career research.”(page 5 line 93-94 of the manuscript without track changes and page 5 line 100, 101 of the manuscript with the track changes). 

4. Much of the statistical jargon used might not be readily understood by the casual reader of PLOS ONE. Perhaps a brief explanation would help the casual reader for some of the topics. For example,

“sampling and snowballing”, “convenience sample” in lines 130-131.

“JvE, KSJ and LG” on line 153

“By-person factor analysis, centroid factor analysis, varimax rotation” line 172-173- is it possible to provide a brief 1-2 sentence explanation describing how these showed a maximum of three factors?

We further elaborated on convenience sample, sampling and snowballing in the participants section:

“As we had no prior information about which career orientations exist among students and which students have these different orientations, a combination of convenience sampling and snowballing was used. As a starting point of the data collection we approached a convenience sample of medical students, i.e. an easy-to-reach group consisting of participants varying in year of medical school and background characteristics that we anticipated to be relevant for capturing the diversity of viewpoints among medical students, as is common in Q-methodology.[23] In order to reach a more diverse group, snowballing was used. This means we asked participants from the convenience sample to suggest consecutive participants, based on their expectations that these other students would have similar or dissimilar career aspirations. The main purpose of our sampling strategy was including a diverse group of medical students that would help us to identify the variety in career orientations.”(page 7 line 142-152 of the manuscript without track changes and page 7 line 152-165 of the manuscript with the track changes) 

We introduced the abbreviations JvE, KSJ and LG by introducing those in the author order line. (page 1 line 2, 3 of both the manuscript with and without track changes). 

We rephrased the analysis part of the method aiming to be more clear with regard to the by-person factor analysis, centroid factor analysis and varimax rotation:

“By-person factor analysis was used to identify distinct career orientations from the individual ranking data.[23] First, a correlation matrix between the rankings of the statements by the participants is computed. Assuming that if two participants rank the 62 statements in a similar way, they have a similar career orientation, factor analysis (i.e., centroid factor extraction followed by varimax rotation [23]) is then applied to identify the main patterns in the ranking data. The selection of the number of patterns (or factors) to extract from the data is based on statistical criteria (i.e., Eigenvalue >1 and a minimum of two participants statistically significantly associated with the pattern) and whether patterns have a coherent interpretation that is also supported by the corresponding qualitative data from the interviews.

For each identified pattern, an idealized ranking of the data is computed. This is a weighted average ranking of the 62 statements, based on the rankings of participants statistically significantly associated with the pattern (p<.0.05) and their correlation coefficient as weight. 

These idealized rankings are then interpreted as distinct career orientations among medical students. Interpretation of each pattern starts by the characterising statements, those with a +5, +4, -4 or -5 score in that pattern, and the distinguishing statements, those with a statistically significant different score (p<0.05) in a pattern as compared to the other patterns. However, a pattern consists of all 62 statements and it is the interrelationship of the many items that ultimately drive our interpretation. 

Finally, the qualitative interview data (i.e. the motivations for their ranking of the statements provided by participants) from participants statistically significantly associated with the pattern (p<.0.05) are used to verify and refine the interpretations. Selected quotes from these qualitative materials are used to substantiate the interpretations of the patterns.” (Page 9 line 194-214 of the manuscript without track changes and page 9-10, line 209- 229 of the manuscript with track changes). 

5. Is it possible to include an example of the paper-based questionnaire (without student answers)? Line 170

The paper-based questionnaire is added in the Appendix. Page 32 of the manuscript without track changes and page 34 of the manuscript with track changes. 

6. Please clarify the use of “participants to identify consecutive participants” (line 134), does this mean that students participating in the study would recruit other students to participate in the study? Would this potentially hurt the diversity of students in the study? Could this potentially explain why 15 out of the 24 subjects were female?

We further elaborated on our sampling strategy: “As we had no prior information about which career orientations exist among students and which students have these different orientations, a combination of convenience sampling and snowballing was used. As a starting point of the data collection we approached a convenience sample of medical students, i.e. an easy-to-reach group consisting of participants varying in year of medical school and background characteristics that we anticipated to be relevant for capturing the diversity of viewpoints among medical students, as is common in Q-methodology.[23] In order to reach a more diverse group, snowballing was used. This means we asked participants from the convenience sample to suggest consecutive participants, based on their expectations that these other students would have similar or dissimilar career aspirations. The main purpose of our sampling strategy was including a diverse group of medical students that would help us to identify the variety in career orientations.” (page 7 line 142-152 of the manuscript without track changes and page 7 line 152-164 of the manuscript with track changes)

With regard to the over-representation of women we added a sentence in the results section: “These twenty-four students were from medical schools at two universities, 9 were male and 15 were female. The mean age of participants was 21.3, with a range of 18-27 years. This gender and age distribution is representative of the total student populations.[25]” (page 10 line 228-229 of the manuscript without track changes and page 12 line 267-271 of the manuscript with track changes)

7. Approximately how many clinical and medical education professionals participated in the creation of the statement set (lines 143-144)?

Three clinical and 3 medical education professionals participated in the creation of the statement set. To be more clear we rephrased the sentence regarding the team involved in the statement set development: “Therefore, a team of clinical and medical education professionals joined forces to develop the statement set.” (page 7 line 160, 161 of the revised manuscript without track changes and on page 8, line 173, 174 of the revised manuscript with the track changes).

8. Is it possible to include a figure describing how a maximum of 3 career orientations could explain 48% of the variance. This is a major part of the study and it would be helpful to help the novice reader understand how the data can be used to show that three different career paths could be obtained from the data.

We are not aware of a method of visualizing this procedure. We clarified the procedure by rephrasing the analysis part of the methods. These changes can be found on page 9 line 194-202 of the manuscript without track changes and page 9,10 and line 209-229 of the manuscript with track changes. 

9. Do the authors have any comment/discussion on how work-life balance was the one career path overwhelmingly chosen by female students?

We further elaborated on this finding in the discussion section: “Remarkably, particularly female participants defined the career orientation work-life balance. This methodology is not suited to make definitive statements about prevalence of a career orientation in certain subgroups. However, one might wonder whether work-life considerations might still be of more importance to female medical students. Notwithstanding, recent research showing the importance of work-life balance for male students,[27] our data suggest that female participants give a higher priority to work-life considerations.” (page 14, line 333-339 of the revised manuscript without track changes and on page 16, line 379-387 of the revised manuscript with the track changes)

10. Some of the discussion can be consolidated and shortened e.g. “Medical students with a work-life balance orientation to their career primarily expressed a desire for a good balance between work and private life” lines 284-286

Thank you for the suggestion. We changed the discussion and this sentence. 

11. Was there a survey to demonstrate that the students felt that the practice benefited their values, needs and motivations? Authors indicated the students felt that they benefited from the exercise, yet if no official survey was done, they should state so. Lines 315-317, and 367-368

As the reviewer points out, we were not clear enough in describing the data underpinning our statement that the students felt that the practice benefits their values, needs and motivations. We elaborated on this by adding the following sentences: “In the interviews following the ranking of the 62 statements, students were asked how they experienced the sorting exercise. All participants recognised that prioritising their values, needs, and motivations in this particular way helped them to get insight in their priorities. In their opinion this would help them in thorough career decision making, thereby suggesting that Q-sorting could be used as a career guidance tool.”Page 15, 16, line 258-362 of the revised manuscript without track changes and page 17, line 409-414 the revised manuscript with the track changes).

12. Table 1 – please label Pattern 1, 2, 3 with regards to the indicated career orientation, e.g. work-life balance

We labeled the patterns with the career orientation names. Page 27, 28 of the revised manuscript without track changes and page 29, 30 of the revised manuscript with the track changes)

13. Is it possible to include additional tables that rank each of the statement for the three career orientations?

For example Career orientation 1: List most important at top, least important at bottom, do for the other career orientations. Table 1 by itself is hard to quickly process by itself.

We added tables with characterising statements in order to enable a better quick overview of our results. These tables can be found on page 29, 30, 31 of the revised manuscript without track changes and on page 31, 32, and 33 of the revised manuscript with the track changes.

We would like to thank the reviewers again for their time and energy invested in reviewing our manuscript. We feel the suggestions of the reviewers improved our manuscript.

Please let us know if any more changes are required. We hope you will accept our paper for publication in PLOS ONE.

---

## [Decision Letter · Decision Letter 1]

11 Mar 2021

Career orientations of medical students: a Q-methodology study

PONE-D-20-29474R1

Dear Dr. Gennissen,

We’re pleased to inform you that your manuscript has been judged scientifically suitable for publication and will be formally accepted for publication once it meets all outstanding technical requirements.

Kind regards,

Sina Safayi, D.V.M., Ph.D.

Academic Editor

PLOS ONE

Reviewers' comments:

Reviewer's Responses to Questions

**Comments to the Author**

1. If the authors have adequately addressed your comments raised in a previous round of review and you feel that this manuscript is now acceptable for publication, you may indicate that here to bypass the “Comments to the Author” section, enter your conflict of interest statement in the “Confidential to Editor” section, and submit your "Accept" recommendation.

Reviewer #1: All comments have been addressed

Reviewer #5: All comments have been addressed

Reviewer #6: (No Response)

2. Is the manuscript technically sound, and do the data support the conclusions?

Reviewer #1: Yes

Reviewer #5: Yes

Reviewer #6: Yes

3. Has the statistical analysis been performed appropriately and rigorously? 

Reviewer #1: Yes

Reviewer #5: Yes

Reviewer #6: Yes

4. Have the authors made all data underlying the findings in their manuscript fully available?

Reviewer #1: Yes

Reviewer #5: Yes

Reviewer #6: Yes

5. Is the manuscript presented in an intelligible fashion and written in standard English?

Reviewer #1: Yes

Reviewer #5: Yes

Reviewer #6: Yes

6. Review Comments to the Author

Reviewer #1: Well done and thanks for your revision and responses - it reads much better and your analysis and conclusions are quite compelling. I look forward to seeing your follow-on studies.

Reviewer #5: I recommend publication based on the improvements to the paper. Thank you to the authors for the time they put in to the study and writing it up.

Reviewer #6: (No Response)

7. PLOS authors have the option to publish the peer review history of their article (what does this mean?). If published, this will include your full peer review and any attached files.

Reviewer #1: No

Reviewer #5: No

Reviewer #6: No

---

## [Editor Report · Acceptance letter]

12 May 2021

PONE-D-20-29474R1 

Career orientations of medical students: a Q-methodology study 

Dear Dr. Gennissen:

I'm pleased to inform you that your manuscript has been deemed suitable for publication in PLOS ONE. Congratulations! Your manuscript is now with our production department. 

Kind regards, 

on behalf of

Dr. Sina Safayi 

Academic Editor

PLOS ONE